# Ovarian Cancer Targeting Phage for In Vivo Near-Infrared Optical Imaging

**DOI:** 10.3390/diagnostics9040183

**Published:** 2019-11-10

**Authors:** Mallika Asar, Jessica Newton-Northup, Susan Deutscher, Mette Soendergaard

**Affiliations:** 1Department of Chemistry, Western Illinois University, 214 Currens Hall, Macomb, IL 61455, USA; m-asar@wiu.edu; 2Department of Biochemistry, University of Missouri, 117 Schweitzer Hall, Columbia, MO 65211, USA; newtonj@missouri.edu (J.N.-N.); deutschers@missouri.edu (S.D.); 3Harry S. Truman Memorial Veterans Hospital, 1 Hospital Drive, Columbia, MO 65201, USA

**Keywords:** phage, imaging, ovarian cancer, molecular targeting, nanoparticles, peptides

## Abstract

Ovarian cancer is often diagnosed at late stages due to current inadequate detection. Therefore, the development of new detection methods of ovarian cancer is needed. This may be achieved by phage nanoparticles that display targeting peptides for optical imaging. Here, two such phage clones are reported. Ovarian cancer binding and specificity of phage clones (pJ18, pJ24) and peptides (J18, J24) were investigated using fluorescent microscopy and modified ELISA. Further, AF680-labeled phage particles were subjected to biodistribution and optical imaging studies in SKOV-3 xenografted mice. Fluorescent microscopy and ELISA of phage and peptides showed significantly increased binding to SKOV-3 cells compared to controls. Additionally, these studies revealed that J18 exhibits specificity for ovarian cancer SKOV-3 and OVCAR-3 cell lines. Further, peptides displayed increased SKOV-3 binding compared to N35 (non-relevant peptide) with EC_50_ values of 22.2 ± 10.6 μM and 29.0 ± 6.9 (mean ± SE), respectively. Biodistribution studies of AF680-labeled phage particles showed tumor uptake after 4 h and excretion through the reticuloendothelial system. Importantly, SKOV-3 tumors were easily localized by optical imaging after 2 h and 4 h and displayed good tumor-to-background contrast. The fluorescent tumor signal intensity was significantly higher for pJ18 compared to wild type (WT) after 2 h.

## 1. Introduction

Ovarian cancer is the most fatal gynecological malignancy, accounting for more than 14,000 deaths annually in the U.S. The majority of patients are diagnosed at stages III and IV, at which point five-year survival rates are 72% and 27%, respectively [1]. However, a more optimistic prognosis is presented to patients diagnosed with local disease, for which five-year survival rates are over 90%. These statistics emphasize the importance of early detection and diagnosis. At present, the standard ovarian cancer detection method is the measurement of glycoprotein CA-125 serum levels [2,3]. However, only 80% of advanced ovarian cancers and 50% of stage I tumors secrete detectable levels of CA-125. Additionally, CA-125 is associated with other conditions such as benign ovarian cysts, endometriosis, and inflammatory disease, and is found in several normal tissues [3,4,5]. Thus, new methods of ovarian cancer detection and diagnosis are needed.

Bacteriophage (phage) is employed in phage display, a high throughput technology utilized to identify peptide and antibody-based ligands [6,7,8]. Phage display has been used to discover novel peptides that target and image cancer biomarkers and cells, including the ErbB-2 receptor, galectin-3, PC-3 prostate carcinoma, and SKOV-3 ovarian adenocarcinoma cells [9,10,11,12,13,14]. While free peptide ligands may be used in targeted cancer imaging, phage as nanoparticles offer advantages in regard to their large size, which allows labeling with numerous fluorophores or radiotracers. Such labeling leads to signal amplification and provides increased sensitivity [15]. Further, phage particles exhibit increased stability and biodistribution compared to single ligands [16]. Finally, phage nanoparticles may be easily genetically modified and produced in large quantities using standard biotechnological approaches, including recombinant DNA technology and propagation in liquid *E. coli* culture. Phage nanoparticles have been utilized as tumor imaging agents of various targets such as Lewis lung carcinoma and prostate cancer in mouse models using optical imaging [13,17]. In fact, phage mediated cancer-targeting coupled with optical imaging, provide additional advantages, such as low toxicity compared to radionuclides. Although the use of fluorophores with emission wavelengths in the visible spectrum is problematic due to tissue depth limitations [18], near-infrared fluorophores (700–900 nm) emit light that penetrates human tissue at a depth of circa 1–3 cm [19,20]. While this depth of tissue penetration prevents imaging of certain tumors, ovarian cancer metastases are most often located along the peritoneal lining, and may, therefore, be imaged using this technique [19].

Here, we report the development of two phage nanoparticles (pJ18 and pJ24) that display 15-mer ovarian cancer-targeting peptides. The phage were previously identified using a two-tier phage display selection against xenografted ovarian tumors in mice. Micropanning experiments revealed that pJ18 and pJ24 exhibited the highest cancer-to-normal cell-binding ratio and these phage were thus selected for further studies [10]. In this study, phage and single peptides were evaluated separately using in vitro cell-based assays and fluorescent microscopy, which revealed affinity in the μmolar range and specificity for ovarian cancer cells. The phage clones were labeled with near-infrared fluorophore Alexa Fluor 680 (AF680) and investigated in regard to tumor targeting and imaging of human ovarian tumors in xenografted nude mice. Phage pJ18 successfully imaged tumors with sufficient tumor-to-background uptake, suggesting that this clone may be used for subsequent utilization in detection and diagnosis of ovarian cancer.

## 2. Materials and Methods

### 2.1. Chemicals and Reagents

Chemicals were purchased from ThermoFisher Scientific (Waltham, MA, USA) unless otherwise stated.

### 2.2. Cell Lines and Cell Culture

The human ovarian adenocarcinoma (SKOV-3), human ovarian adenocarcinoma (OVCAR-3), human melanoma (MDA-MB-435), and human embryonic kidney (HEK293) cell lines were obtained from American Type Tissue Culture (Manassas, VA, USA). SKOV-3 cells were maintained in McCoy’s 5A supplemented with 10% fetal bovine serum (FBS) and 50 μg/mL gentamicin at 37 °C in 5% CO_2_. All other cell lines were maintained in RPMI 1640 (custom) supplemented with 10% fetal bovine serum (FBS), 2 mM L-glutamine, and 1.7 µM insulin at 37 °C in 5% CO_2_.

### 2.3. Animals and Handling

Female 4–6-week-old nude nu/nu mice (*n* = 2; Harlan, Indianapolis, IN, USA) were injected subcutaneously (shoulder) with 1 × 10^7^ SKOV-3 cells under anesthesia (3.5% isoflurane, Baxter Healthcare Corp., Deerfield, IL, USA), and tumors (1 cm) were established over 8 weeks. After optical imaging studies, the animals were sacrificed by cervical dislocation. The NIH Guidelines for the Care and Use of Laboratory Animals and the Policy and Procedures for Animal Research of the Harry S. Truman Veterans Memorial Hospital were followed, and all animal studies were approved by the Institutional Animal Care and Use Committee (IACUC; 6823) at Harry S. Truman Veterans Memorial Hospital.

### 2.4. Phage

A linear 15-amino acid fUSE5 phage display library (a gift from Dr. George P. Smith [21]) was previously employed in a two-tier selection protocol against SKOV-3 cells, as previously reported to select and identify phage pJ18 and pJ24 [10]. Phage were amplified in *E. coli* K91 Blue Kan (K91BK) and isolated as previously described [22]. In brief, *E. coli* K91BK colonies infected with phage were propagated overnight in NZY medium with 100 μg/mL kanamycin and 20 μg/mL tetracycline at 37 °C and 225 rpm. The culture was then centrifuged at 5000× *g* for 15 min, and the phage were precipitated from the supernatant using 0.15 volume polyethylene glycol (PEG)/NaCl overnight at 4 °C. Next, phage were collected by centrifugation (8000× *g*, 15 min), resuspended in tris buffered saline (TBS), and precipitated once more using 0.15 volume PEG/NaCl. The virions were then immediately collected by centrifugation (8000× *g*, 15 min), and then resuspended in 1 mL TBS. The concentration of virions was determined by measuring the absorbance at 269 nm and 320 nm as described previously [22].

### 2.5. Peptides

Peptides (J18: RSLWSDFYASASRGP, J24: RRLPHLMPFEGSVFL, and N35: APGPTFLD) were synthesized with a biotin group linked to a GSG-amino acid spacer at the NH_2_-terminus (biotin-GSG-peptide) by solid-phase Fmoc chemistry (396 multiple peptide synthesizer, Advanced Chem Tech, Louisville, KY, USA). Peptides were purified on a C18 column (Novapack Reverse Phase, Waters, Milford, MA, USA) by reverse-phase high-pressure liquid chromatography (RP-HPLC; Beckmann Coulter System Gold HPLC, Beckmann Coulter, Fullerton, CA, USA), and identified by electrospray ionization mass spectrometry (Mass Consortium Corp, San Diego, CA, USA). Peptides were lyophilized and stored at −20 °C.

### 2.6. Fluorescent Microscopy

Binding of phage pJ18, pJ24, wild type (WT), and dimethyl sulfoxide (DMSO; vehicle) to SKOV-3 cells were evaluated using fluorescent microscopy. SKOV-3 cells were maintained on microscope chamber slides (Nest Scientific, Rahway, NJ, USA) to 80% confluency. Cell medium was aspirated and washed thrice with TBS. Cells were fixed using 0.2% Triton X-100 and 10% formalin in TBS and washed extensively. Phage pJ18, pJ24, and WT (10^11^ V/mL) in 1% bovine serum albumin (BSA) TBS were incubated with biotinylated anti-Fd filamentous phage antibody (Fitzgerald, Acton, MA, USA) and streptavidin-fluorescein isothiocyanate (FITC) conjugate for 1 h. The phage complex was then added to SKOV-3 cells for 1 h. Cells were washed repeatedly with TBS and mounted using ProLong™ Diamond Antifade Mountant with 4′,6-diamidino-2-phenylindole (DAPI). Bound phage were detected using a Keyence BZ-X800 fluorescent microscope (Keyence, Itasca, IL, USA).

Cells (SKOV-3, OVCAR-3, MDA-MB-435, and HEK293) were grown to 80% confluency on chamber slides (Lab-Tek, Rochester, NY, USA). Growth medium was aspirated and 10 µM biotinylated J18 or J24 in phosphate buffered saline (PBS) were added for 1 h at 37 °C. Next, the slides were washed (PBS), fixed with 10% formalin, and then washed (PBS) repeatedly. The cells were blocked (10% FBS, 0.3 M glycine, 0.01% Tween-20, PBS) for 1 h, and bound peptides were probed with a mouse monoclonal anti-biotin antibody labeled with Cy3 (Sigma Aldrich, St. Louis, MO, USA) for 1 h. The slides were washed (PBS, 0.05% Tween-20), and bound peptides were detected using an epifluorescent Nikon T1-SM inverted microscope (Nikon, Melville, NY, USA).

The fluorescent intensity per cell was quantified by determining the corrected total cell fluorescence (CTCF) using ImageJ software (v1.52a, National Institute of Health, Bethesda, MD, USA) [23]. The mean fluorescent intensity per cell (region of interest; ROI) was adjusted for the mean background (non-cell) fluorescent intensity.

### 2.7. Modified ELISA

The binding ability of phage pJ18, pJ24, and WT, and biotinylated peptides J18, J24, and a non-relevant peptide N35 (negative control) to SKOV-3 cells were analyzed using an in vitro assay. Cells were grown in 96-well tissue culture plates (TPP, Trasadingen, Switzerland) to 80% confluency.

For phage, the cell medium was aspirated, and the cells were fixed using 0.2% Triton X-100 and 10% formalin. Cells were then washed with TBS, and 10^9^ or 10^10^ V/mL pJ18, pJ24, or WT phage (1% BSA; TBS) was pre-incubated with biotinylated anti-Fd filamentous phage antibody (Fitzgerald, Acton, MA, USA) and streptavidin-horse radish peroxidase (HRP)-conjugate, and added to the cells for 1 h. The cells were washed (TBS) and incubated with HRP substrate 2, 2′-azino-bis(3-ethylbenzothiazoline-6-sulphonic acid) (ABTS) for 30 min. Phage binding was detected by measuring the absorbance at 405 nm on a SpectraMax 250 microplate reader (Molecular Devices, San Jose, CA, USA).

To determine peptide binding, 10 nM to 300 µM of biotinylated J18, J24, or N35 (non-relevant peptide) in PBS were incubated with the cells for 1 h at 37 °C, after which the plates were washed repeatedly (1% BSA, PBS) and fixed with 10% formalin. After extensive washing, the cells were blocked (10% FBS, 0.3 M glycine, 0.05% Tween-20 in PBS), and the bound peptides were probed by incubation with streptavidin-conjugated HRP for 1 h. Next, the plate was washed (PBS, 0.05% Tween-20), and ABTS was added and allowed to develop for 20 min. Bound peptides were detected by measuring the absorbance at 405 nm using a µ Quant Universal Microplate Spectrophotometer (Bio-Tek Instruments, Winooski, VT, USA), and the half-maximal effective concentration (EC_50_) was calculated using GraphPad Prism software (vs. 8.0, San Diego, CA, USA).

### 2.8. Fluorescent Labeling of Phage

The near-infrared fluorophore Alexa Fluor 680 (AF680; carboxylic acid, succinimidyl ester 5-isomer; Invitrogen, Carlsbad, CA, USA) was dissolved in 2% dimethyl sulfoxide (DMSO) and incubated with phage (0.29 mM coat protein VIII) in 0.5 M Na_3_ citrate, 0.1 M NaHCO_3_ pH 8.5 for 4 h at room temperature in the dark. The reaction was terminated overnight at 4 °C by addition of 270 mM ethanolamine, pH 9. Excess hydrolyzed AF680 was removed by dialyzing labeled phage against TBS, pH 7.5 (Slide-A-Lyzer cassette, 10 kDa molecular weight cutoff, ThermoFisher Scientific, Rockford, IL, USA).

### 2.9. Biodistribution and Near-Infrared Optical Imaging of AF680-Labeled Phage

AF680-labeled pJ18, pJ24 or WT (10^12^ transducing units; TU) were injected into the tail-vein of SKOV-3 xenografted mice and allowed to circulate. For biodistribution experiments, the mice were sacrificed after 4 h, perfused with PBS, and tumors, organs, and tissues were excised. For optical imaging experiments, the mice were under gas anesthesia (3.5% isoflurane, Baxter Healthcare Corp. Deerfield, IL, USA), and fluorescence reflectance images were obtained before injection (0 h), and 2 h and 4 h after injection. Images were attained, and the fluorescent intensity was measured using a Xenogen IVIS 200 System and analyzed by Living Image software (vs. 4.3.1, Perkin Elmer, Waltham, MA, USA) [23].

### 2.10. Statistical Analysis

Two-way analysis of variance (ANOVA) and Student’s *t*-test were performed to analyze the statistical significance of the fluorescent quantification, modified phage ELISA, and fluorescent intensity for the optical imaging experiment using Prism GraphPad Software (vs. 8.0, San Diego, CA, USA). A *p*-value of ≤0.05 was considered significant.

## 3. Results

### 3.1. Fluorescent Microscopy

Phage and peptide binding to various cell lines was assessed using fluorescent microscopy. Phage pJ18, pJ24, and WT (10^11^ V/mL) were incubated with SKOV-3 cells, and the binding was gauged by biotinylated anti-Fd filamentous phage antibody and FITC. Results showed that pJ18 and pJ24 exhibited significantly increased binding to SKOV-3 cells compared to WT and DMSO (Figure 1).

Individual peptides, J18 and J24, were synthesized with an N-terminal GSG-spacer and a biotin group and employed in fluorescent microscopy studies to further evaluate the binding characteristics and to verify that the interaction was mediated by the displayed peptides and not by inherent phage proteins. The binding of J18 and J24 were assessed against both cancerous (SKOV-3, OVCAR-3, MDA-MB-435) and non-cancerous (HEK293) cell lines. In brief, peptides were incubated with each cell line and probed with a mouse monoclonal anti-biotin antibody labeled with Cy3. These studies revealed that peptide J18 displayed significantly increased binding for ovarian cancer SKOV-3 and OVCAR-3 cell lines while exhibiting minimal affinity for breast cancer MDA-MB-435 and embryonic kidney HEK293 cells. Peptide J24 showed significantly higher binding to SKOV-3 cells and little to no binding to other cell lines (Figure 2).

### 3.2. Modified ELISA

In order to further evaluate the SKOV-3 binding properties of both phage and peptides, these were subjected to in vitro cell binding assays. For analysis of phage binding, 10^9^ or 10^10^ V/mL pJ18, pJ24, or WT were incubated with SKOV-3 cells and probed by biotinylated anti-Fd filamentous phage antibody and streptavidin-HRP conjugate. The phage–cell interaction was measured spectrophotometrically at 405 nm after the addition of ABTS. Phage pJ18 and pJ24 (10^9^ V/mL) failed to show increased binding to SKOV-3 cells compared to WT. However, at 10^10^ V/mL pJ18 and pJ24 both exhibited significantly increased SKOV-3 binding compared to WT, indicating that the interaction is mediated by the displayed peptides, rather than by inherent non-specific binding of the virion (Figure 3).

To further validate that the displayed peptides mediate the binding interaction, and not the intrinsic phage proteins, a cell-based dose-response assay was carried out. For this, SKOV-3 cells were incubated with different concentrations of biotinylated peptides (10 nM to 300 µM), after which binding was probed by streptavidin-HRP and detected spectrophotometrically at 405 nm after addition of ABTS (Figure 4). Peptides J18 and J24 displayed increased binding to SKOV-3 cells in comparison to a non-relevant peptide (negative control; N35) and exhibited EC_50_ values of 22.2 ± 10.6 μM and 29.0 ± 6.9 (mean ± SE), respectively.

### 3.3. Phage Biodistribution Studies

The biodistribution of AF680-labeled phage clones pJ18, pJ24, and WT was investigated in female nude mice bearing xenografted SKOV-3 tumors. Phage were injected into the tail-vein of the animals; after 4 h the mice were sacrificed by cervical dislocation, and the fluorescent intensity in the tumors and normal tissues was measured (Figure 5). Both phage clones pJ18 and pJ24 exhibited fluorescent tumor uptake after 4 h; however, the levels were not significantly higher compared to WT. pJ24 exhibited significantly higher uptake in lung, kidneys, and liver compared to WT phage. The reticuloendothelial system was the major route of excretion for all phage (pJ18, pJ24, and WT), which was apparent from increased uptake primarily in the liver, lungs, and spleen. However, the kidneys also showed increased fluorescent intensity for pJ18 and pJ24, indicating that renal clearance is a minor excretion route. Uptake of phage in the heart, muscle, and brain did not differ between the clones.

### 3.4. Near-Infrared Optical Imaging

Near-infrared optical imaging of nude mice carrying xenografted SKOV-3 tumors was performed to investigate the imaging properties of phage clones pJ18 and pJ24 (Figure 6a). AF680-labeled phage (pJ18, pJ24, or WT) were tail-vein injected into the animals and circulated for up to 4 h. The mice were imaged before the injection (0 h) to determine autofluorescence, as well as 2 h and 4 h post-injection. The obtained images revealed that the SKOV-3 tumors were easily localized and exhibited sufficient tumor-to-background contrast (Figure 6a). The fluorescent tumor signal intensity of each ROI was significantly higher in comparison to WT for pJ18 at 2 h post-injection. The tumor uptake decreased after 4 h for all phage clones. Although the signal intensity remained higher for pJ18 compared to WT, the difference was no longer significant (Figure 6b).

## 4. Discussion

Bacteriophage offer advantages as molecular imaging agents, in that they may be easily produced in *E. coli* and genetically modified to display various ligands. Additionally, phage may be utilized in both the development and delivery of such modalities. First, these nanoparticles may be employed in phage display technology to select and identify peptide-based ligands. Second, once such ligands have been validated, peptide-displaying phage may be used as biological nanoparticles that mediate binding to a molecular target, and that may at the same time carry various imaging agents. Phage particles have previously been successfully labeled with fluorophores, radiotracers, and iron, and used in optical [9,13] and radio imaging [24], as well as magnetic resonance imaging of molecular targets [25]. The large size of Fd filamentous phage also allows labeling with up to hundreds of imaging molecules per virion, typically on coat protein VIII (cpVIII), thereby providing signal amplification [15]. The spatial separation of the fluorophore on cpVIII and display of the foreign peptides on genetically modified coat protein III (cpIII) diminish any steric hindrance between the groups, thereby minimally affecting peptide-mediated target binding [13]. However, phage often exhibit non-specific hydrophobic binding [26,27,28]. For this reason, it is vital to validate that the displayed peptide mediates phage binding, and not inherent coat proteins. To do so, the modified ELISA and fluorescent microscopy experiments were carried out using peptide-displaying and WT phage, as well as single peptides.

Fluorescent microscopy was used to investigate phage binding to ovarian adenocarcinoma (SKOV-3) cells, and revealed that both clones, pJ18 and pJ24, displayed increased binding compared to WT. These results indicate that SKOV-3 binding of pJ18 and pJ24 is mediated by the respectively displayed peptides, as only minimal binding was observed for a WT phage, which is void of a foreign peptide. In order to further validate this observation, fluorescent microscopy studies were carried out using individual biotinylated peptides (J18 and J24) against both cancerous (SKOV-3, OVCAR-3, MDA-MB-435) and non-cancerous (HEK293) cell lines. The results showed that peptide J18 exhibited affinity for ovarian cancer cell lines (SKOV-3 and OVCAR-3), but only minimal binding to breast cancer (MDA-MB-435) and embryonic kidney (HEK293) cells. To the contrary, peptide J24 only exhibited binding to SKOV-3 cells, with negligible binding to other cell lines. Taken together, these results suggest that the selected peptides J18 and J24 bind ovarian cancer-specific antigens, and may, as a result, be utilized as ovarian carcinoma imaging agents. Further, the different binding characteristics of J18 and J24 for SKOV-3 and OVCAR-3 cells indicate that the peptides may target different antigens. SKOV-3 cells are known to exhibit more invasive potential compared to OVCAR-3, and to express higher levels of epidermal growth factor receptor (EGFR) [29,30,31]. The higher affinity of J18 for SKOV-3 may, therefore, suggest that the peptides bind an antigen with high and low expression profiles in the two cell lines, respectively, such as EGFR. The minimal binding to MDA-MB-435 cells, which lack EGFR expression, further supports this notion [32]. However, additional studies are needed to validate this observation. The high SKOV-3-to-OVCAR-3 binding ratio of J24 may also indicate that the peptide targets an antigen that is involved in invasiveness and that is comparatively overexpressed in the more aggressive SKOV-3 cell line [30,31]. For these reasons, peptide J18 may be binding to an antigen present in a wider array of ovarian cancer types and could, therefore, be useful in more broad applications, while J24 may be utilized to provide a more specialized targeting strategy.

As pJ18 and pJ24 was selected against SKOV-3 cells in a two-tier *in vivo* phage display selection [10], binding of the phage to this cell line was further investigated using a modified ELISA. The results revealed that at a concentration of 10^9^ V/mL pJ18, pJ24, and WT showed similar binding to SKOV-3 cells. Nevertheless, at 10^10^ V/mL pJ18 and pJ24, both exhibited significantly increased binding compared to WT. The latter results (10^10^ V/mL) suggest, as previously noted, that the phage–SKOV-3 cell interaction is mediated by the displayed peptides, and not by non-specific binding of the virion. However, these results also indicate that 10^9^ V/mL of pJ18 and pJ24 is insufficient to mediate detectable cell binding.

Phage from the fUSE5 phage display library express one to five peptides on genetically modified cpIII [33], and thus a phage concentration of 10^9^ V/mL correlates to a peptide concentration of 1.7–8.3 pM. Therefore, to further evaluate the binding interaction, a cell-based dose-response assay using peptide concentrations (10 nM to 300 µM) above those used in the phage-based assay (1.7–8.3 pM) was performed. Both peptides J18 and J24 displayed saturation binding kinetics to SKOV-3 cells with EC_50_ values of 22.2 ± 10.6 μM and 29.0 ± 6.9 (mean ± SE), respectively. In comparison, a non-relevant peptide (N35; negative control) showed only minimal binding, from which it was not possible to calculate an EC_50_ value. The EC_50_ values of J18 and J24 are consistent with previously reported peptides that were identified by phage display technology [9,34]. In fact, phage that display peptides with EC_50_ values in the micromolar range have previously been used in optical imaging of xenografted tumors in mice [9]. Finally, the saturation binding kinetics further indicate that the peptide–cell interactions are specific [35], which confirms the previous notion that phage pJ18 and pJ24 binding is mediated by the displayed peptides, and not by non-specific phage proteins.

In order to evaluate the pharmacokinetics of phage pJ18 and pJ24, these were labeled with the near-infrared fluorophore AF680 and employed in biodistribution studies. After 4 h, fluorescent tumor uptake was detected for all phage clones (pJ18, pJ24, and WT). However, for pJ18 and pJ24 the tumor uptake was not significantly higher compared to WT. The low tumor levels may be a result of rapid pharmacokinetics and could suggest that optimal uptake occurs at earlier time points. Most other organs (lung, spleen, kidney, bladder, skin, and liver) showed increased fluorescent levels. In fact, pJ24 showed significantly higher uptake in the lungs, liver, and kidneys indicating elevated background binding compared to WT. Comparing the hydrophobicity of pJ18 and pJ24 according to the Wimley–White hydrophobicity scale [36], it is evident that pJ24 is comparatively more hydrophobic. Hydrophobicity is associated with the binding of nanoparticles to plasma proteins [37,38], and even slight increases in hydrophobicity have been shown to influence and extend biodistribution times [39,40]. For all phage, the liver exhibited increased fluorescent uptake compared to other organs, indicating that the phage were excreted through the reticuloendothelial system (RES). This observation was expected, as nanoparticles are cleared primarily through the RES [41,42], which was further confirmed by elevated uptake by the spleen and lungs. However, the kidneys also displayed increased fluorescent intensity, especially for pJ24. These results suggest that the kidneys may be a minor excretion route for both pJ18 and pJ24, but not for WT phage, which showed only minimal uptake in these organs. Previous biodistribution studies of phage have shown similar results regarding reticuloendothelial excretion [9,13,43,44,45]. While the large size and biochemical properties of nanoparticles, including phage, are known to greatly affect biodistribution and excretion [46], the display of peptides have shown to partly re-direct the clearance [13]. Taking these observations into account, the pharmacokinetic results in this study may indicate that peptides J18 and J24 are steering the excretion system towards the kidneys. Phage uptake in other organs (heart, muscle, and brain) was comparatively lower and did not differ between the clones. The low uptakes were possibly caused by the tight junctions that connect heart and muscle cells and challenge extravasation of large particles into these tissues. Additionally, the blood–brain barrier excludes the vast majority of particles above 400 kDa, which significantly limits the ability of phage to enter the brain [47].

Near-infrared optical imaging of SKOV-3 xenografted nude mice was carried out to evaluate the tumor imaging capabilities of AF680-labeled pJ18, pJ24, and WT. The images revealed that the tumors were easily localized and showed satisfactory tumor-to-background contrast. Peak tumor fluorescent intensity was obtained after 2 h for all phage clones, and at this time point, the signal intensity was significantly higher for pJ18 compared to WT. In correlation to the biodistribution study, the tumor uptake of pJ18 and pJ24 after 4 h was similar to that of WT. These observations may be due to high uptake and clearance rates, which for pJ18 is further supported by the similarly low levels in other tissues at this time point. However, the imaging study showed that pJ24 failed to demonstrate increased tumor uptake at any time point. This observation may indicate that pJ24 exhibits optimal tumor uptake at a time point before 2 h or after 4 h of circulation. The surface charge and displayed functional groups of nanoparticles are known to influence both cellular uptake and excretion; consequently, the small difference in positive charge at physiological pH of peptide J18 (charge +1 at pH 7) and J24 (charge +1.1 at pH 7) may be the reason behind the observed differences [41].

Overall, this optical imaging study shows that the peptides, J18, and J24, influence the binding characteristics of the phage particles. Phage clone pJ18 enables specific targeting and imaging of SKOV-3 tumors in xenografted mice. Thus, this phage nanoparticle may be utilized in future studies to improve current detection strategies of ovarian cancer.

Peptide displaying phage, pJ18 and pJ24, were previously selected using phage display technology for binding to human ovarian adenocarcinoma SKOV-3 cells [10]. Here, fluorescent microscopy and modified ELISA showed increased and specific binding of phage, pJ18 and pJ24, and peptides, J18 and J24, to ovarian cancer cells compared to controls. In vivo studies revealed successful tumor targeting of AF680-labeled pJ18, enabling easy tumor visualization. In conclusion, this study demonstrates the ability of pJ18 to target and image xenografted ovarian tumors *in vivo*. Therefore, this phage clone may be utilized in further studies as an ovarian cancer-targeting nanoparticle to improve the detection and diagnosis of the disease.

## Figures and Tables

**Figure 1 diagnostics-09-00183-f001:**
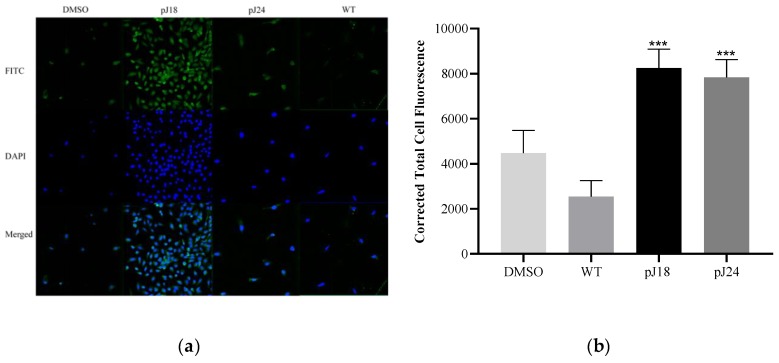
Fluorescent microscopy of phage binding to SKOV 3 cells. (**a**) Cells were incubated with 10^11^ V/mL pJ18, pJ24, wild type (WT), or dimethyl sulfoxide (DMSO) (vehicle) and probed by a biotinylated anti-Fd filamentous phage antibody and streptavidin-fluorescein isothiocyanate (FITC). Cells were further stained with ProLong™ Diamond Antifade Mountant with 4′,6-diamidino-2-phenylindole (DAPI) and detected using a Keyence BZ X800 fluorescent microscope (Keyence, Itasca, IL, USA). (**b**) The corrected total cell fluorescence was determined using ImageJ software, and showed that pJ18 and pJ24 exhibited higher fluorescence compared to WT and DMSO. Phage pJ18 and pJ24 also showed significantly higher fluorescence compared to DMSO (not shown). A *p*-value of <0.05 was considered significant. *** *p* < 0.001. Green (FITC-phage); blue (DAPI).

**Figure 2 diagnostics-09-00183-f002:**
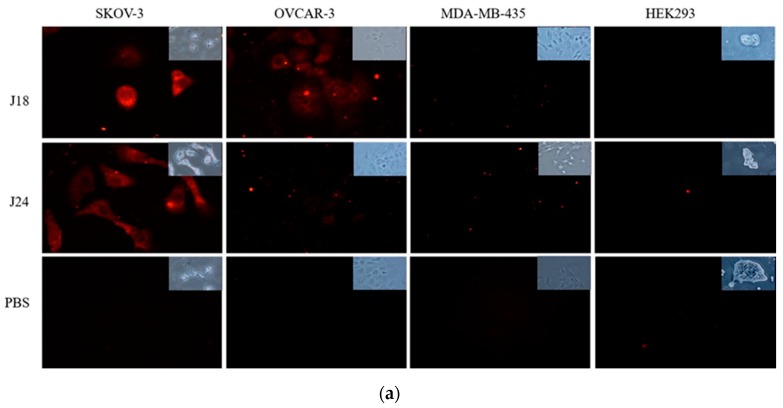
(**a**) Fluorescent microscopy of peptide binding to SKOV-3, OVCAR-3, MDA-MB-435, and HEK293 cells. Cells were incubated with 10 µM biotinylated J18, J24, or PBS for 1 h at 37 °C. Bound peptides were probed by a mouse monoclonal anti-biotin antibody and detected using an epifluorescent Nikon T1-SM inverted microscope (Nikon, Melville, NY, USA). (**b**) The corrected total cell fluorescence was determined using ImageJ software. Results showed that peptide J18 displayed higher fluorescence compared to PBS (vehicle) for SKOV-3 and OVCAR-3 cells. Peptide J24 demonstrated higher fluorescence to SKOV-3 cells. A *p*-value of < 0.05 was considered significant. **** *p* < 0.0001. Red (cy3-peptides).

**Figure 3 diagnostics-09-00183-f003:**
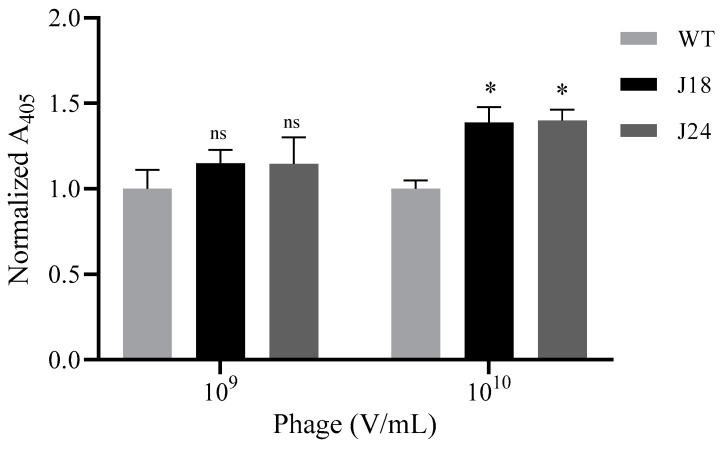
Binding of phage to ovarian carcinoma (SKOV-3) cells. Cells were grown to 80% confluency on 96-well plates, fixed using 0.2% Triton X-100 and 10% formalin, and then incubated with 10^9^ or 10^10^ V/mL pJ18, pJ24, or WT phage. Phage binding was probed by biotinylated anti-Fd filamentous phage antibody and streptavidin-HRP, and detected spectrophotometrically at 405 nM after addition of 2, 2′-azino-bis(3-ethylbenzothiazoline-6-sulphonic acid) (ABTS). Binding of phage pJ18 and pJ24 to SKOV-3 cells was significantly higher at 10^10^ V/mL compared to WT. Measurements were performed on a SpectraMax 250 microplate reader (Molecular Devices, San Jose, CA, USA). A *p*-value of <0.05 was considered significant. * *p* < 0.05.

**Figure 4 diagnostics-09-00183-f004:**
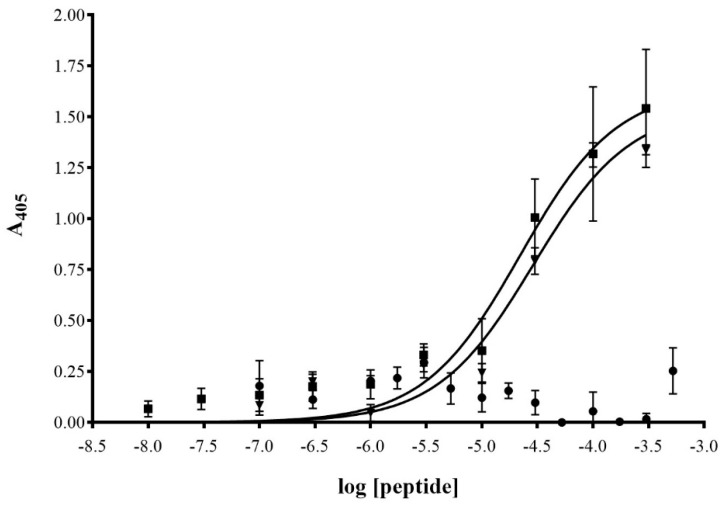
Binding properties of biotinylated peptides J18, J24, and N35 to ovarian carcinoma (SKOV-3) cells. Cells were grown to 80% confluency on 96-well plates and then incubated with varying concentrations (10 nM to 300 μM) of J18 (■), J24 (▼), and N35 (•) for 1 h at 37 °C. Plates were washed with PBS and fixed with 10% formalin. Biotinylated peptides were detected by HRP-conjugated streptavidin and measurement of absorbance at 405 nM after the addition of HRP-substrate ABTS. EC50 values for J18 and J24 were calculated to be 22.2 ± 10.6 μM, and 29.0 ± 6.9 μM (mean ± SE), respectively. Measurements were performed on a µ Quant Universal Microplate Spectrophotometer (Winooski, VT, USA).

**Figure 5 diagnostics-09-00183-f005:**
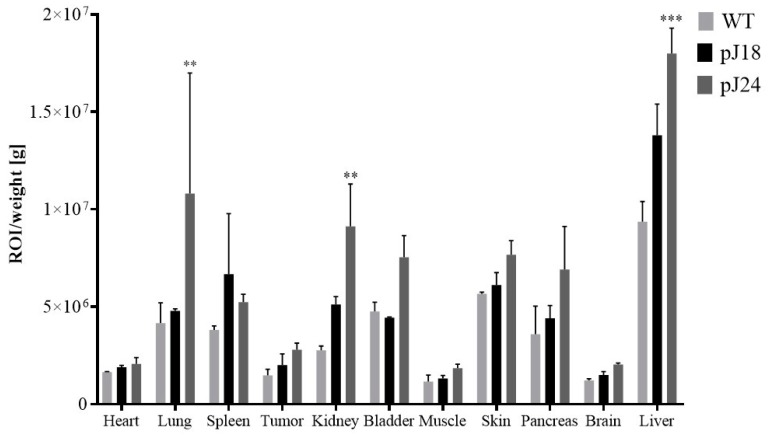
Biodistribution of AF680-labeled pJ18, pJ24, and WT phage in SKOV-3 xenografted nude female mice. Mice (*n* = 2) were injected with 10^12^ virions of phage. After 4 h the animals were sacrificed, and the organs, tissues, and tumors were excised and weighed. The fluorescent intensity was measured using a Xenogen IVIS 200 System. Fluorescent uptake was normalized to the weight of each organ and reported as fluorescent intensity per gram. Fluorescent intensity/g was significantly higher for pJ24 in the lungs, kidneys, and liver as compared to WT. A *p*-value of <0.05 was considered significant. ** *p* < 0.01, *** *p* < 0.001.

**Figure 6 diagnostics-09-00183-f006:**
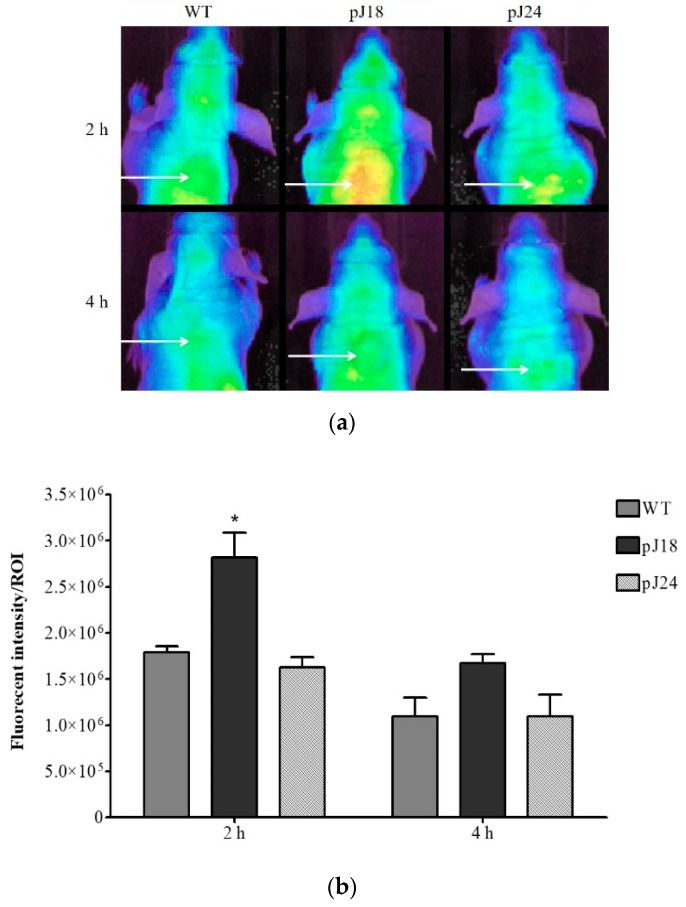
In vivo optical imaging of SKOV-3 xenografted tumors in female nude mice (*n* = 2) using AF680-labeled pJ18, pJ24, and WT phage. The mice were injected with 10^12^ virions of phage and imaged under anesthesia at 2 h and 4 h post-injection. (**a**) Fluorescence reflectance images of SKOV-3 xenografted mice. A white arrow indicates tumor location. (**b**) Quantification of fluorescent signal intensity of the region of interest (ROI). Fluorescence reflectance images were obtained using a Xenogen IVIS 200 System, and the fluorescent signal intensity was quantified using ImageJ software. Fluorescent intensity/ROI was significantly higher for pJ18 after 2 h compared to WT. A *p*-value of <0.05 was considered significant. * *p* < 0.05.

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
