# Peer review of "Ovarian Cancer Targeting Phage for In Vivo Near-Infrared Optical Imaging"

_diagnostics, 2019, doi:10.3390/diagnostics9040183_

Round 1

Reviewer 1 Report

Comments:

Include the AUP approval number from Harry S. Truman Veterans Memorial Hospital.

The selection of phage pJ18 and pJ24 may be justified well.

Section 2.4: The Phage amplification in E. coli K91 Blue Kan (K91BK) and further isolation need to be explained with details.

Did the authors performed micropanning assay to identify teh specific clones for the cells?

The authors performed the biodistribution of AF680-Labeled Phage using IR imaging. However the biodistribution within the tissue samples were analysed (reproductive organs)?

Did the authors tested the phage coat protein VIII?

Author Response

Include the AUP approval number from Harry S. Truman Veterans Memorial Hospital.

The selection of phage pJ18 and pJ24 may be justified well.

The following was added to the last paragraph of the introduction: The phage were previously identified using a two-tier phage display selection against xenografted ovarian tumors in mice. Micropanning experiments revealed that pJ18 and pJ24 exhibited the highest cancer-to-normal cell-binding ratio and were thus selected for further studies [10].

Section 2.4: The Phage amplification in E. coli K91 Blue Kan (K91BK) and further isolation need to be explained with details.

The following was added to section 2.4: In brief, E. coli K91BK colonies infected with phage were propagated overnight in NZY medium with 100 μg/mL kanamycin and 20 μg/mL tetracycline at 37°C and 225 rpm. The culture was then centrifuged at 5000 xg for 15 min, and the phage were precipitated from the supernatant using 0.15 volume polyethylene glycol (PEG)/NaCl overnight at 4°C. Next, phage were collected by centrifugation (8000 xg, 15 min), resuspended in tris buffered saline (TBS), and precipitated once more using 0.15 volume PEG/NaCl. The virions were then immediately collected by centrifugation (8000 xg, 15 min), and then resuspended in 1 mL TBS. The concentration of virions was determined by measuring the absorbance at 269 nm and 320 nm as described previously [22].

Did the authors performed micropanning assay to identify teh specific clones for the cells?

Micropanning was performed previously by the authors (Soendergaard et al., 2014. Am J Nucl Med Mol Imaging 2014, 4, 561-570). The results showed that pJ18 and pJ24 exhibit the highest cancer-to-normal cell-binding ratios. 

The authors performed the biodistribution of AF680-Labeled Phage using IR imaging. However the biodistribution within the tissue samples were analysed (reproductive organs)?

The reviewer makes an excellent point to investigate the murine reproductive organs. However, these were not analyzed, since the phage were originally selected against human ovarian cancer cells, and analyzed in human ovarian tumors in xenografted mice. Unfortunately, it is not possible for us to re-analyze the reproductive organs of the mice.

Did the authors tested the phage coat protein VIII?

The selected peptides are displayed on coat protein III and not cpVIII (fUSE5 library). It is possible to display peptides on cpVIII; however, that was not chosen for this study. Mostly, we chose cpIII expression to avoid "false" binding that often occurs on cpVIII due to the avidity effect. The cpVIII was chosen for labeling of fluorescent molecules since this provides the ability to label with hundreds of fluorophores per virion. We used WT phage as a control to ensure that the binding to cancer cells was not mediated by the fluorescent molecules, but by the displayed peptide.

Reviewer 2 Report

In this study, Mallika et al. developed two phage nanoparticles (pJ18 and pJ24) that display 15-mer ovarian cancer-targeting peptides for the detection and diagnosis of ovarian cancer. The authors confirmed the specificity of these phage nanoparticles by in vitro cell-based ELISA assays and fluorescent microscopy. Of note, this study demonstrated the ability of the pJ18 phase clone to specifically target and image xenografted SKOV-3 tumors in vivo.

In general, this manuscript is well-written that provides a good rationale for the development of phage nanoparticles for better ovarian tumor visualization. There are some overall issues, however, in the logical flow of the experimentation as well as the specific scientific details which are described below.

Major:

All the in vitro studies need to be done independently at least thrice to show the consistency. However, the authors only showed the representative images and did not provide the quantifications and significances of each experimental group in some results. Also, the descriptions of figure legends are not cleared or misplaced. Figure 1: Please explain why the densities of SKOV-3 cells in each experiment are different. Do the cells incubated with pJ18 phages also showed an increased cell proliferation? What does the DMSO group mean? Does the DMSO group also incubate with phage nanoparticles? If so, which phage nanoparticles were incubated with the cells treated with DMSO? The DMSO group seems to show a similar binding ability with the pJ24 group in SKOV-3 cells. Also, the scale bar and quantification of signals are missing. The labels of groups shown in the figure should be pJ18 and pJ24. Figure 2: It is informative that pJ18 phage nanoparticles showed specific binding in both SKOV-3 and OVCAR-3 ovarian cancer cell lines but not in melanoma MDA-MB-435 or HEK293 cells. However, the human normal ovarian cells (e.g., HS-832) may be a better negative control since the authors want to apply this for ovarian cancer diagnosis. Also, what is the difference between pJ18 and pJ24 phages as they displayed different binding abilities in ovarian cancer cells? The description of figure legend is incorrect and seems to be Figure 3. Again, the scale bar and quantification of signals are missing, and the labels of groups shown in the figure should be pJ18 and pJ24. Figure 3: The same experiments should also be done in OVCAR3. The description of figure legend is incorrect and seems to be Figure 2.

Minor:

It would be great if the authors update the reference regarding cancer statistics in 2019 (Siegel et al., CA Cancer J Clin, 2019). Please use a consistent format for all the figures. Figure legends: The number of independent biological replicates should be indicated and at least 3 independent biological replicates should be performed. Also, please clarify the groups with significance (*), are they all compared to WT? Figure 4: The labels of each group are missing in the J18 (■), J24 (▼), WT (●)?

Author Response

All the in vitro studies need to be done independently at least thrice to show the consistency. However, the authors only showed the representative images and did not provide the quantifications and significances of each experimental group in some results.

Also, the descriptions of figure legends are not cleared or misplaced.

Figure legends have been updated:

Figure 2: Fluorescent microscopy of peptide binding to SKOV-3, OVCAR-3, MDA-MB-435 and HEK293 cells. Cells were incubated with 10 µM biotinylated J18, J24, or PBS for 1 h at 37°C. Bound peptides were probed by a mouse monoclonal anti-biotin antibody and detected using an epifluorescent Nikon T1-SM inverted microscope (Nikon, Melville, NY).

Figure 3: Binding of phage to ovarian carcinoma (SKOV-3) cells. Cells were grown to 80% confluency on 96-well plates, fixed using 0.2% Triton X-100 and 10% formalin, and then incubated with 109 or 1010 V/mL pJ18, pJ24, or WT phage. Phage binding was probed by biotinylated anti-Fd filamentous phage antibody and streptavidin-HRP, and detected spectrophotometrically at 405 nM after addition ABTS. Binding of phage pJ18 and pJ24 to SKOV-3 cells was significantly higher (p < 0.05) at 1010 V/mL compared to WT. Measurements were performed on a SpectraMax 250 microplate reader (Molecular Devices, San Jose, CA).

Figure 1: Please explain why the densities of SKOV-3 cells in each experiment are different. Do the cells incubated with pJ18 phages also showed an increased cell proliferation?

We do not know why the densities of the SKOV-3 cells are different. This may reflect random imaging of more or less populated colonies. Unfortunately, we cannot reimage these slides, since they were done during an equipment demonstration. However, we do know that J18 does not increase cell proliferation. In fact, J18 is cytotoxic as determined by an MTT assay (unpublished data to be submitted later this year). 

What does the DMSO group mean? Does the DMSO group also incubate with phage nanoparticles? If so, which phage nanoparticles were incubated with the cells treated with DMSO? The DMSO group seems to show a similar binding ability with the pJ24 group in SKOV-3 cells.

DMSO is dimethyl sulfoxide, which is used as the vehicle in the experiment. The phage are diluted in DMSO, and thus DMSO is the vehicle control. 

Also, the scale bar and quantification of signals are missing.

These images were acquired during an equipment demonstration session (Keyence BZ X800). Unfortunately, we no longer have access to the software to add a scale bar.

The labels of groups shown in the figure should be pJ18 and pJ24.

The figure labels have been updated to pJ18 and pJ24.

Figure 2:

The figure 2 figure legend has been changed to: Fluorescent microscopy of peptide binding to SKOV-3, OVCAR-3, MDA-MB-435 and HEK293 cells. Cells were incubated with 10 µM biotinylated J18, J24, or PBS for 1 h at 37°C. Bound peptides were probed by a mouse monoclonal anti-biotin antibody and detected using an epifluorescent Nikon T1-SM inverted microscope (Nikon, Melville, NY).

It is informative that pJ18 phage nanoparticles showed specific binding in both SKOV-3 and OVCAR-3 ovarian cancer cell lines but not in melanoma MDA-MB-435 or HEK293 cells. However, the human normal ovarian cells (e.g., HS-832) may be a better negative control since the authors want to apply this for ovarian cancer diagnosis.

Yes, the HS832 cell line would have been a better negative control. Unfortunately, we lost access to this cell line as we and the Cell and Immunobiology Core at the University of Missouri failed to propagate these after just a few doublings. This is often the case for normal cells. We tried for months to propagate the cells, but eventually they were all lost. 

Also, what is the difference between pJ18 and pJ24 phages as they displayed different binding abilities in ovarian cancer cells?

The differences lie in the sequence of their displayed peptides. It is currently unknown which molecular target (protein, carbohydrate) that these peptides bind to. However, it is very possible that they target two different markers, as they exhibit different binding to SKOV-3 and OVCAR-3 cells. These differences are also discussed in the discussion.

The description of figure legend is incorrect and seems to be Figure 3.

Figure 3 has been updated to:

Binding of phage to ovarian carcinoma (SKOV-3) cells. Cells were grown to 80% confluency on 96-well plates, fixed using 0.2% Triton X-100 and 10% formalin, and then incubated with 109 or 1010 V/mL pJ18, pJ24, or WT phage. Phage binding was probed by biotinylated anti-Fd filamentous phage antibody and streptavidin-HRP, and detected spectrophotometrically at 405 nM after addition ABTS. Binding of phage pJ18 and pJ24 to SKOV-3 cells was significantly higher (p < 0.05) at 1010 V/mL compared to WT. Measurements were performed on a SpectraMax 250 microplate reader (Molecular Devices, San Jose, CA).

Again, the scale bar and quantification of signals are missing, and the labels of groups shown in the figure should be pJ18 and pJ24.

As described earlier, we, unfortunately, don't have access to the software anymore. Therefore, we cannot add the scale bars. The labels have been corrected.

Figure 3: The same experiments should also be done in OVCAR3.

This is a valid point. However, we currently don't have access to the OVCAR-3 cell line in the WIU laboratory. 

The description of figure legend is incorrect and seems to be Figure 2. This has been changed to match the correct figures. See changes above.

Minor:

It would be great if the authors update the reference regarding cancer statistics in 2019 (Siegel et al., CA Cancer J Clin, 2019).

This has been updated.

Please use a consistent format for all the figures.

Figures have been updated to be consistent. 

Figure legends: The number of independent biological replicates should be indicated and at least 3 independent biological replicates should be performed.

Binding studies using modified ELISA were all done in at least triplicates. Animal studies were done with an n=2. It is currently nor possible to add any more animal experiments to the study.

Also, please clarify the groups with significance (*), are they all compared to WT?

This has been clarified in the appropriate figure legends. pJ18 and pJ24 phage were compared to WT, and a p-value of <0.05 was considered significant. The degrees of significance have been updated in the figure legends.

Figure 4: The labels of each group are missing in the J18 (■), J24 (▼), WT (●)? 

Figure 4 has been updated to include a label for each peptide: 

Binding properties of biotinylated peptides J18, J24 and J30 to ovarian carcinoma (SKOV-3) cells. Cells were grown to 80% confluency on 96-well plates, and then incubated with varying concentrations (10 nM to 300 μM) of J18 (■), J24 (▼) N35 (•) for 1 h at 37°C. Plates were washed with PBS and fixed with 10% formalin. Biotinylated peptides were detected by HRP-conjugated streptavidin and measurement of absorbance at 405 nM after addition of HRP-substrate ABTS. EC50 values for J18 andJ24 were calculated to be 22.2 ± 10.6 μM, and 29.0 ± 6.9 μM (mean ± SE), respectively. Measurements were performed on a µ Quant Universal Microplate Spectrophotometer (Winooski, VT).

N35 (•) is a negative control peptide that does not bind to any of the cell lines that we have worked with.

Reviewer 3 Report

In this manuscript, the two ovarian cancer targeting phage clones (pJ18, pJ24) were investigated in vitro and in vivo by the authors. The manuscript is well-organized and interesting.

Although I am not an expert on this aspect, but I do have several concerns that the authors need to address in order to publish a well-read article.

In the ms, the author stated that the reticuloendothelial system was the major route of excretion for all phage, then how can we tell if the tumor exist or not in this system when this phage technique applied to clinical use? From the results, we understand that peptide displaying phage, pJ18 and pJ24, were selected against human ovarian cancer SKOV-3 cells and peptide J18 displayed binding for ovarian cancer SKOV-3 and OVCAR-3 cell lines while peptide J24 showed only binding to SKOV-3 cells. So the question is which phage is better? If the answer is pJ24 for its binding specificity, then we all know the ovarian cancer is a heterogeneous disease, how can we solve this problem? Conversely, if it is pJ18, how to solve the specificity? Line 20, the statement in abstract is inconsistent with the result of figure 2. Figure 6b showed that pJ24 failed to display the tumor location at any time point and the authors though that pJ24 may exhibit optimal tumor uptake at a time point before 2 h or after 4 h or circulation. Please set an experiment to prove it, because I think it is very important to demonstrate that pJ24 can function normally in vivo.

Author Response

In the ms, the author stated that the reticuloendothelial system was the major route of excretion for all phage, then how can we tell if the tumor exist or not in this system when this phage technique applied to clinical use?

This is a valid point, and a question that we often receive. In clinical use, it would be possible to have patients empty their bladder prior to imaging. This would reduce background imaging in the pelvic region. Further, humans are larger than mice, which facilitates differentiation between organs and tissues. 

From the results, we understand that peptide displaying phage, pJ18 and pJ24, were selected against human ovarian cancer SKOV-3 cells and peptide J18 displayed binding for ovarian cancer SKOV-3 and OVCAR-3 cell lines while peptide J24 showed only binding to SKOV-3 cells. So the question is which phage is better? If the answer is pJ24 for its binding specificity, then we all know the ovarian cancer is a heterogeneous disease, how can we solve this problem?

This is also a valid point. For diagnosis, an imaging agent should ideally be able to detect multiple sub-types of disease to diagnose as many patients as possible. Also, it is important that the imaging agent exhibits sufficient affinity for the target to be adequately sensitive. Thus, pJ18/J18 is the best diagnostic imaging agent as concluded in the discussion.

Conversely, if it is pJ18, how to solve the specificity?

The specificity issue does not need to be solved (see answer above), since pJ18/J18 is specific for ovarian cancer cell lines (at least for the two tested). However, it would be a problem if the phage/peptide also bound to other cancer or normal cells.

Line 20, the statement in abstract is inconsistent with the result of figure 2.

Yes, this was an error. The abstract has been updated.

Figure 6b showed that pJ24 failed to display the tumor location at any time point and the authors though that pJ24 may exhibit optimal tumor uptake at a time point before 2 h or after 4 h or circulation. Please set an experiment to prove it, because I think it is very important to demonstrate that pJ24 can function normally in vivo.

We agree with this statement. It is possible that our experiment didn't encompass the optimal time points for pJ24. The biodistribution and imaging experiments were carried out on the same mice. Therefore, it was not possible to analyze the biodistribution at 2 h (this requires sacrifice of the animals). We may plan to carry out additional studies in the future to determine the optimal biodistribution and imaging time points of pJ24.

Round 2

Reviewer 1 Report

The authors need to include the AUP approval number from Harry S. Truman Veterans Memorial Hospital to use the 4–6-week-old female nude nu/nu mice.

Fig 6B. The authors may cross-check the statistics again.

Author Response

The authors need to include the AUP approval number from Harry S. Truman Veterans Memorial Hospital to use the 4–6-week-old female nude nu/nu mice.

The following has been updated in the Materials and Methods:

“The NIH Guidelines for the Care and Use of Laboratory Animals and the Policy and Procedures for Animal Research of the Harry S. Truman Veterans Memorial Hospital were followed, and all animal studies were approved by the Institutional Animal Care and Use Committee (IACUC; 6823) at Harry S. Truman Veterans Memorial Hospital”.

Both institutions, the University of Missouri and Harry S. Truman Veterans Memorial Hospital are approved to handle experimental animals. Animal welfare assurance number (MU): A3394-01. Animal welfare assurance number (VA): A3157-01.

Fig 6B. The authors may cross-check the statistics again.

                The statistics of Figure 6b was re-analyzed using GraphPad Prism (8.0.0). The raw data for Figure 6 b is shown below:

2 h

Phage

ROI

Average

STD

CV

WT

1.83E+06

1.75E+06

1.79E+06

5.52E+04

3.1

#18

3.01E+06

2.63E+06

2.82E+06

2.72E+05

9.7

#24

1.50E+06

1.76E+06

1.63E+06

1.80E+05

11.0

4 h

Phage

ROI

Average

STD

CV

WT

1.26E+06

9.35E+05

1.10E+06

2.33E+05

21.2

#18

1.50E+06

1.85E+06

1.67E+06

2.47E+05

14.8

#24

8.48E+05

1.34E+06

1.10E+06

3.50E+05

31.9

                An unpaired two-tailed Students t-test was done for pairwise comparison of WT and pJ18 (2 h), WT and pJ24 (2 h), WT and pJ18 (4 h), and WT and pJ24 (4 h). Results showed that WT and pJ18 (2 h) was significantly different (p=0.0337). A p-value <0.05 was considered significant.

Reviewer 2 Report

Figure 1: In section 2.6 Fluorescent Microscopy, the methods read "SKOV-3 cells were maintained on microscope chamber slides (Nest Scientific, Rahway, NJ) to 80% confluency.......The phage complex was then added to SKOV-3 cells for 1 h". Hence, the cell densities should be similar in different treatments if the cell were seeded to 80% confluency and only incubated with phages for 1 h.

Also, the authors claimed that "However, we do know that J18 does not increase cell proliferation. In fact, J18 is cytotoxic as determined by an MTT assay (unpublished data to be submitted later this year)." However, the results shown in Figure 1 demonstrated greater proliferation in cells treated with pJ18 compared to either WT or pJ24. It seems to be the opposite of what they claimed.

Further, according to the authors' reply, they have done these experiments at least triplicate. So it won't be hard to find original images of these experiments in different fields. Therefore, it would be better to provide similar cell densities of immunofluorescence images and quantification to avoid confusion. 

The other question is why the cells treated with DMSO without incubated with FITC conjugated-phage complex (DMSO group) still showed the FITC signals. Aren't they supposed to be FITC negative?

Round 3

Reviewer 2 Report

The manuscript is much improved and most of my comments have been addressed satisfactorily in the revision